# Cold Exposure Exacerbates Cardiac Dysfunction in a Model of Heart Failure with Preserved Ejection Fraction in Male and Female C57Bl/6J Mice

**DOI:** 10.3390/biomedicines13081900

**Published:** 2025-08-04

**Authors:** Sara-Ève Thibodeau, Marie-Lune Legros, Emylie-Ann Labbé, Élisabeth Walsh-Wilkinson, Audrey Morin-Grandmont, Sarra Beji, Marie Arsenault, Alexandre Caron, Jacques Couet

**Affiliations:** 1Department of Medicine, Faculty of Medicine, Laval University, Quebec City, QC G1V 0A6, Canada; sara-eve.thibodeau@criucpq.ulaval.ca (S.-È.T.); marie-lune.legros.1@ulaval.ca (M.-L.L.); emylie-ann.labbe@criucpq.ulaval.ca (E.-A.L.); elisabeth.walsh-wilkinson.1@ulaval.ca (É.W.-W.); audrey.morin-grandmont@criucpq.ulaval.ca (A.M.-G.); marie.arsenault@criucpq.ulaval.ca (M.A.); 2Valvulopathies Research Group, Research Center of the University Institute of Cardiology and Pneumology of Quebec, Laval University, Quebec City, QC G1V 4G5, Canada; 3Faculty of Pharmacy, Laval University, Quebec City, QC G1V 0A6, Canada; sarra.beji.1@ulaval.ca (S.B.); alexandre.caron@criucpq.ulaval.ca (A.C.); 4Research Center of the University Institute of Cardiology and Pneumology of Quebec, Laval University, Quebec City, QC G1V 0A6, Canada

**Keywords:** housing temperature, cold stress, heart failure, brown fat, cardiac hypertrophy, HFpEF

## Abstract

**Background**: Standard room temperature housing (~22 °C) represents a stress for laboratory mice, resulting in an increased metabolic rate, calorie consumption, heart rate, and catecholamine levels compared to thermoneutral conditions (29–32 °C). Using a recently established two-hit model of heart failure with preserved ejection fraction (HFpEF) (Angiotensin II + High-fat diet for 28 days; MHS), we investigated how housing temperature modulates cardiac remodelling and function in male and female C57Bl/6J mice. **Methods**: Using the MHS mouse model, we investigated cardiac remodelling and function in 8-week-old C57BL/6J mice of both sexes housed at 10 °C, 22 °C, and 30 °C for four weeks. Control mice were analyzed in parallel. Before the MHS, the animals were allowed to acclimate for a week before the MHS started. **Results**: Mice housed at 10 °C consumed more food and had increased fat mass compared to those at 22 °C or 30 °C. This was accompanied by increased heart weight, stroke volume, heart rate, and cardiac output. Mice housed at 22 °C and 30 °C were similar for these cardiac parameters. Following MHS, mice at 10 °C and 22 °C developed marked cardiac hypertrophy, whereas thermoneutral housing attenuated this response and reduced left atrial enlargement. Cold-exposed females showed more diastolic dysfunction after MHS (increased E’ wave, E/E’, and isovolumetric relaxation time) than those at 22 °C or 30 °C. Ejection fraction and cardiac output declined significantly at 10 °C after MHS but were preserved at 22 °C and 30 °C in females. **Conclusions**: Cold housing exacerbates cardiac dysfunction in mice subjected to HFpEF-inducing stress, with pronounced effects in females. In contrast, thermoneutrality limits the cardiac hypertrophic response.

## 1. Introduction

Several murine models of heart failure with preserved ejection fraction (HFpEF) have been developed in recent years, most commonly using C57Bl/6 mice. These models typically combined hypertensive stress (e.g., L-NAME, Angiotensin II, DOCA-Salt) with metabolic alterations (e.g., high-fat or Western diets). Some of these models have been studied in males, females, or both. Combining multiple stresses (or hits) better reflects the multifactorial nature of HFpEF in human patients [1,2,3,4].

One key physiological difference between humans and mice is their ability to regulate body temperature, also known as thermoregulatory capacity. Thermoneutrality in mice is estimated to range between 26 °C and 32 °C [5,6,7]. Mice are typically housed at a temperature that creates a cold-stressful environment (22–23 °C). Under these conditions, mice display elevated catecholamine levels, increased energy expenditure through adaptive thermogenesis, and increased brown adipose tissue (BAT) activity [5,6,7,8,9]. Chronically elevated catecholamines are known contributors to the development and progression of heart failure, suggesting that cold stress could accelerate cardiac dysfunction in susceptible mouse models. A recent study using a two-hit HFpEF model revealed that the onset of diastolic dysfunction occurred later in mice housed at 30 °C compared to those housed at 23 °C. However, housing temperature was not a differentiating factor if the stress persisted [10].

BAT activation, often triggered by cold exposure, has been proposed to have cardioprotective effects through the release of endocrine factors known as “batokines”. These include FGF21, neuregulin 4, 12,13-diHOME, and BAT-derived microRNAs, all of which have been implicated in modulating cardiac or vascular function in the context of cardiovascular diseases [11].

Thus, while cold exposure activates BAT thermogenesis and batokine secretion, it also induces physiological stress that may negatively impact cardiovascular health [12]. However, the impact of cold exposure on the pathophysiology of HFpEF remains unclear.

To address this, we utilized a recently developed two-hit mouse model of HFpEF [13] to investigate the effect of housing temperature on cardiac structure and function. Male and female C57Bl6/J mice were housed for five weeks at 10 °C (cold), 22 °C (room temperature), or 30 °C (thermoneutrality). Half of the animals were subjected to the MHS protocol, consisting of an Angiotensin II (AngII) continuous infusion combined with a high-fat diet for 28 days (MHS) [13].

We report that housing mice at 10 °C is associated with BAT activation, cardiac hypertrophy, left ventricular (LV) remodelling, and increased cardiac output. In MHS-treated mice, cold exposure further exacerbated diastolic dysfunction. In contrast, thermoneutral housing decreased LV mass (as assessed by echocardiography and cardiomyocyte size) and attenuated the cardiac hypertrophic response. Furthermore, the expression of pathological gene markers was less pronounced at thermoneutrality compared to cold stress conditions.

## 2. Materials and Methods

### 2.1. Animals

Seven-week-old male and female C57Bl6/J mice were purchased from Jackson Laboratory (Bar Harbor, ME, USA). Mice were housed on a 12 h light–12 h dark cycle with free access to food and water. Mice were individually housed at three different temperatures: thermoneutrality (30 °C), room temperature (22 °C), and cold temperature (10 °C). The protocol was approved by the Université Laval’s animal protection committee and followed the recommendations of the Canadian Council on Laboratory Animal Care (approval numbers 2023-1549 and 2023-1550, dated February 2023). Mice were randomly assigned to the various experimental groups (n = 6 per group). Health and behaviour were monitored daily by experienced technicians, and body weight was measured weekly. Three intact male mice died in the MHS group.

### 2.2. Metabolic and Hypertensive Stress (MHS) Protocol

As described previously [13], mice in the MHS group were implanted with a subcutaneous osmotic minipump (Alzet #1004) delivering angiotensin II (AngII; 1.5 mg/kg/day) (Sigma, Mississauga, ON, Canada) for 28 days. In parallel, mice were fed a high-fat diet (HFD: 60% calories; Research Diets Cat. #D12492). Control animals were fed a standard chow and no AngII infusion.

### 2.3. Echocardiography

Cardiac function was assessed by echocardiography under isoflurane anesthesia, as described previously [14,15].

### 2.4. Body Composition

After euthanasia and osmotic pump removal, whole-body composition was assessed using a Bruker’s Minispec Whole Body Composition Analyzer (Billerica, MA, USA)

### 2.5. Myocardial Fibrosis Evaluation

Myocardial samples were frozen in OCT, sliced in serial sections (10 μm thick), and stained with Picrosirius Red to assess the percentage of interstitial fibrosis. The formula used to calculate the percentage of interstitial fibrosis was: ((% Fibrosis)/(% Fibrosis + % Tissue)) × 100 [16].

### 2.6. Cardiomyocyte Cross-Sectional Area

The cardiomyocyte cross-sectional area (CSA) was visualized with immunofluorescent wheat germ agglutinin FITC (Sigma) staining as previously described [16].

### 2.7. RNA Isolation and Quantitative Real-Time Polymerase Chain Reaction

As described previously, total RNA was extracted from LV tissue [13]. Quantitative RT-PCR was used to quantify LV gene expression in at least six animals per group. Cyclophilin A (*Ppia*) was the control “housekeeping” gene for studying cardiac genes, and RPL13 for brown fat genes. The primers used are listed in Table 1.

### 2.8. BAT Histology

BAT samples were preserved in a 4% paraformaldehyde solution for 48 h and transferred to 70% ethanol. The specimens were paraffin-embedded, cut into 4 μm sections, and stained with hematoxylin and eosin.

### 2.9. UCP1 Immunohistochemistry

Myocardial samples were fixed in a solution of acetone/methanol (60:40), then washed in Tris-buffered saline (TBS) and treated with sodium citrate at 95 °C. For the staining itself, washes were made in TBS. Incubations were performed in 3% hydrogen peroxide in methanol, followed by blocking in TBS containing 0.3% Triton X-100, 5% BSA, and 5% Goat serum. The primary antibody (Abcam, Toronto, ON, Canada; #155117) was diluted (1:250) in blocking solution and incubated overnight at 4 °C. The second antibody (Cell Signalling Technology, Boston, MA, USA; #7074S; horseradish peroxidase-coupled goat anti-rabbit, 1:400) was incubated for 30 min at room temperature. A DAB substrate kit (Abcam) was used for colour revelation, and hematoxylin/eosin was used as a counterstain.

### 2.10. Statistical Analysis

All data are expressed as mean ± standard error of the mean (SEM). Outliers were removed using the ROUT test with a Q of 1% with Prism. Intergroup comparisons were conducted using Student′s T-test using GraphPad Prism 10.4 (GraphPad Software Inc., La Jolla, CA, USA). Comparisons among more than two groups were analyzed using a two-way ANOVA and the Holm–Sidak post hoc test. *p* < 0.05 was considered statistically significant.

## 3. Results

### 3.1. Different Housing Temperatures Result in Sex-Specific Adaptations in C57Bl6/J Mice

As illustrated in Figure 1A, C57Bl6/J mice were individually housed at three different temperatures—10 °C, 22 °C, or 30 °C—for five weeks. Half of the mice, starting seven days later, received AngII and the high-fat diet (HFD) for four weeks (MHS). The other half remained untreated.

After five weeks, female mice at 10 °C had a similar body weight to those housed at 22 °C or 30 °C (Figure 1B). Their body composition was studied after euthanasia and showed a higher proportion of fat tissue than the two other groups (Figure 1C,D). In males, cold exposure resulted in higher body weight, increased fat content, and slower growth (Figure 1E). Lean mass was reduced in males at 30 °C. As expected, food consumption was highest in mice at 10 °C and lowest in those at 30 °C (Figure 1F).

### 3.2. Housing Temperature Affects Cardiac Morphology and Function

Heart weight indexed to tibial length was highest in cold-housed mice (Figure 2A). Left atrial weight was lower in females housed at thermoneutrality, whereas it was increased in males at 10 °C compared to the two other groups (Figure 2B). Lung weight (wet: Figure 2C and dry: Figure 2D) in mice housed at 22 °C fell between those of animals at 10 °C (higher) and 30 °C (lighter).

We then studied the mouse cardiac morphology and function by echocardiography the day before euthanasia. Using B-mode imaging, we measured LV internal end-diastolic volume (EDV; Figure 2E), LV stroke volume (SV; Figure 2F), the heart rate (HR; Figure 2G), and the resulting cardiac output (CO; Figure 2G). In females, cold exposure was associated with increased EDV, SV, and cardiac output. In males, a similar situation was observed (Table 2). All these parameters remained unchanged between mice kept at standard housing temperature (22 °C) and those maintained at thermoneutrality.

As illustrated in Figure 3, housing at warmer temperatures was associated with smaller LV diastolic and systolic areas, resulting in smaller volumes.

Using M-mode imaging, we observed that the LV internal diastolic diameter (EDD; Figure 4A) was increased in females and that LV walls (Figure 4B) were thicker. Relative LV wall thickness (RWT; Figure 4C) remained unchanged, but estimated LV mass (Figure 4D) was increased. In males, EDD was unchanged, but LV walls were thicker, and LV mass was increased at 10 °C.

As illustrated in Figure 4E, the long-axis sections of the LVs show that the hearts of animals housed in the cold were larger. Cardiomyocyte cross-sectional areas (CSAs) were also larger in these mice (Figure 4F,G). CSA was smaller in mice at thermoneutrality compared to the two other groups.

### 3.3. Thermoneutrality (30 °C) Reduces the Cardiac Hypertrophic Response to MHS

Half of the mice, after one week of acclimatization, were implanted with a mini-osmotic pump delivering a continuous AngII infusion for four weeks and fed a high-fat diet (MHS; Figure 1A). As illustrated in Figure 5A, heart weight was significantly increased after the MHS in both female and male mice, except for mice housed at thermoneutrality. Left atrial enlargement, a feature of HFpEF [2], was present in all groups, but left atrial weight gain after MHS was significantly lower in mice at 30 °C (Figure 5B). CSA was the largest in MHS animals housed in the cold (Figure 5C,D). Myocardial fibrosis was increased in MHS animals and did not differ based on the housing temperature (Figure 5E–G).

### 3.4. Cold Exacerbates Diastolic Dysfunction After MHS and Causes a Loss of Ejection Fraction

When assessed using echocardiography, diastolic function parameters were not markedly altered by the housing temperature (Figure 6A–D). MHS reduced E wave velocities in most groups (Figure 6E). E’ wave velocity was increased only in females at 10 °C and all three male MHS groups compared to controls (Figure 6F). This resulted in an abnormal E/E’ ratio in females housed in the cold (Figure 6G). The isovolumetric relaxation time (IVRT) was increased in male and female mice housed at 10 °C (Figure 6H), as was the left atrial diameter (Figure 6I). A similar observation was also made for the isovolumetric contraction time (IVCT) in mice housed at cold temperatures (Figure 6J).

This longer IVCT in MHS mice at 10 °C suggested that systolic function may have been altered. We used M-mode echocardiography imaging to measure LV wall thickness and internal diameters. As expected, MHS increased LV wall thickness in all groups (Table 2), resulting in an increased relative wall thickness, an index of concentric LV remodelling and increased LV mass. Using bidimensional (B-mode) echocardiography, we observed that MHS decreased LV volumes in males, not females. Stroke volume and cardiac output were reduced in males after MHS. This was the case in females only at 10 °C. Ejection fraction was significantly reduced after MHS in mice housed at 10 °C (Figure 7), but not in the groups housed at 22 °C and 30 °C.

### 3.5. Housing Temperature Modulates Myocardial Hypertrophy and Fibrosis Marker Genes After MHS

We measured the expression levels of several myocardial hypertrophy or fibrosis marker genes to determine if they showed differences in their modulation related to the housing temperature in MHS mice. As illustrated in Figure 8A,B, MHS increased LV gene expression of natriuretic peptides, *Nppa* (atrial) and *Nppb* (brain). For *Nppa* and *Nppb*, this increase was more substantial in groups housed in the cold.

Procollagens 1 and 3 (*Col1a1* and *Col3a1*) gene expression was increased after MHS (Figure 8C,D). In both females and males, MHS did not significantly increase the expression levels of these two genes in mice housed at 22 °C; however, it did so in mice housed at 10 °C and 30 °C. Periostin (*Postn*) and Thrombospondin 4 (*Thbs4*), marker genes of extracellular matrix synthesis and remodelling, increased their expression levels after MHS (Figure 8E,F). This increase was more pronounced in females housed in the cold, but not in males.

### 3.6. Cold Stress Activates BAT, and MHS Induces BAT Browning at Thermoneutrality

At the time of euthanasia, we collected the scapular brown fat depot, a thermogenic organ. As illustrated in Figure 9A, BAT weight was increased in animals housed at 10 °C compared to mice at 22 °C and 30 °C. BAT weight was higher in males at thermoneutrality than at room temperature. Representative pictures of the collected fat depot are illustrated in Figure 9B. Hematoxylin–eosin-stained BAT sections show that thermoneutrality was associated with larger cell fat droplets in male and female mice (Figure 9C). Interestingly, the MHS induced significant histological changes in the BAT of mice housed at 30 °C, similar to those observed in groups housed in cold environments (Figure 9D).

Housing at 10 °C was associated with larger BAT adipocytes (fewer nuclei per microscope field) compared to room temperature (Figure 9E). The MHS stress reduced adipocyte size in both male and female mice. Uncoupling protein 1 (UCP1) is a marker of BAT thermogenesis, and, as expected, its gene expression correlated with housing temperatures, being more expressed in cold-stressed animals (Figure 9F). The MHS did not modulate *Ucp1* gene expression in cold-stressed groups but increased it in mice at thermoneutrality (Figure 9G). Figure 9H shows UCP1 labelling in the BAT of male mice at 10 °C and 22 °C, and undetectable levels at thermoneutrality.

Food consumption was higher in control groups housed at 10 °C compared to those housed at 22 °C or 30 °C (Figure 1F). The HFD contained more calories per gram than the standard diet (5.24 kcal/g vs. 3.82 kcal/g). For mice at 10 °C and 30 °C, the HFD was not associated with higher levels of consumed calories than standard diet-fed mice. Males at 10 °C even consumed less. Only at 22 °C did mice consume more calories with the HFD. (Figure 10).

## 4. Discussion

Most preclinical studies in mice are performed at standard housing temperatures (~22 °C), which represent a cold stress for these animals [5,6,7,8,9,10,17]. Cold exposure increases energy expenditure, food consumption, and locomotion, while individual housing removes opportunities for huddling-based thermoregulation, adding further physiological stress.

Compared to thermoneutrality, cold housing is known to activate the sympathetic nervous system (SNS) and the renin–angiotensin–aldosterone (RAAS) system in mice. Circulating catecholamines are also increased by cold exposure [18,19] as well as plasma renin activity and AngII formation [20,21]. Nitric oxide production, a central vasodilator that controls blood pressure and endothelial function, is reduced by cold exposure via decreased endothelial nitric oxide synthase (eNOS) expression [22]. Evidence also exists that endothelin-1 levels, a potent vasoconstrictor, increase in response to cold exposure [23]. These adaptations to cold could cause increased hypertension (CIH) and heart rate [18,19]. We did not observe changes in heart rate related to cold exposure, as this parameter was recorded during echocardiography exams conducted in mice lying on a heated mat (39 °C).

Continuous recording of blood pressure and heart rate by telemetry would provide insight into the effects of cold and MHS on our mice. In a recent study, thermoneutrality was shown to reduce basal heart rate and increase heart rate variability in mice compared to mice housed at 22 °C [24] It is thus likely that the combination of cold exposure with the MHS increased this parameter in our mice.

Since CIH can cause cardiac hypertrophy, it is plausible that inhibiting the SNS or RAAS systems would block cold-induced cardiac hypertrophy (CICH). Although cold-induced hypertension can be reduced using β-blockers or RAAS inhibitors, this did not translate into blocking CICH [12].

Cold exposure has been associated with an increase in circulating thyroid hormone levels, which are known to activate thermogenesis [25,26]. Interestingly, blockade of thyroid hormone action has been shown to reduce CICH in rats [27]. Since thyroid hormones are known for their involvement in postnatal cardiac growth, a form of physiological CH [28], it is plausible that they, in part, control the development of CICH.

More recently, it was reported that increased oxidative stress could partly modulate CICH. Blocking the expression of the gp91phox-containing NADPH oxidase (Nox2), which is selectively expressed by endothelial cells, using RNA interference, inhibited CICH in rats. This NADPH oxidase is a significant source of oxygen radical generation in the arterial wall [29]. Endothelin-1 inhibition using RNA interference was also shown to inhibit CICH [30]. This also highlights the potential roles of myocardial endothelial cells in the development of CICH and underscores the need to investigate the potential impact of vascular cold adaptations on heart function.

Although some authors consider CICH as a pathological phenotype [12,31,32], others view it as a physiological adaptation to this stress. Increased myocardial capillarity, lipid uptake, and reversibility after stress removal, along with enhanced metabolic capacity, indicate physiological hypertrophy [33]. These authors identify the capacity of a reversal toward a normal phenotype as a hallmark of physiological CH, along with the absence of activation of a myocardial gene profile associated with pathological hypertrophy.

Our results suggest that the continuous demand for cold adaptation can stretch the animal’s capacity, especially the myocardium, to respond to an additional pathological stress, such as the MHS, and exacerbate the severity of the induced phenotype. A similar observation was made in mice, showing that pressure overload-induced cardiac hypertrophy (induced by transverse aortic constriction, or TAC) was accentuated by cold housing [34].

We observed that although CICH in animals housed at 10 °C for five weeks was not associated with changes in the expression of the pathological hypertrophy and fibrosis marker genes we measured, when the MHS was applied, the heart of both male and female mice showed exacerbated cardiac dysfunction compared to mice at thermoneutrality and at ambient room temperature. Our previous studies using the MHS model have shown that males are more prone to lose LV ejection fraction than females. Recovery after MHS cessation was accompanied by a small but significant decline in EF in males, but not in females [13]. Old mice (20 months) exposed to the MHS also resulted in an EF decrease in males, whereas it was maintained in females [35]. Cold exposure at 10 °C had a similar impact on ejection fraction in both sexes. This is unlike many mouse models of cardiac disease, which usually show that females are more resilient than males [36].

This did not appear to be related to exaggerated myocardial fibrosis, as we did not observe higher levels in mice exposed to cold temperatures compared to groups housed at warmer temperatures. We did not evaluate whether cold was associated with an increased presence of immune cells in the myocardium. Since pathological cardiac hypertrophy is often associated with the recruitment of immune cells in the myocardium [37], it is possible that cold exposure could exacerbate this phenomenon.

Both male and female mice after MHS were unable to maintain cardiac output. Cold exposure resulted in larger LV volumes and increased cardiac output. Thus, the LV had to dilate to accommodate this physiological volume overload. The MHS pushes the LV to remodel concentrically. This occurs in the opposite way for the body, which needs to cope with a cold environment, and suggests that the cardiac work imposed on the LV could result in myocyte loss and a decrease in systolic function, in addition to the usual diastolic dysfunction associated with this HFpEF model.

Other signs of a pathological phenotype related to the cold exposure include increased left atrial size and/or mass, lungs’ wet and dry weights, and dilated LV. This phenotype may only reflect a physiological volume overload induced by the need for a higher cardiac output for thermogenesis. On the other hand, the hearts of animals housed at 10 °C were more susceptible to pathological stressors than those housed at higher temperatures.

Adaptations to the cold were similar in male and female mice (increased food consumption and body fat content, dilated LV, higher cardiac output), but several differences were observed. Male mice increased their body weight, whereas females did not, and body growth (as measured by tibial length) slowed in males, but not in females. It would be interesting to investigate whether sex steroids can explain these differences and those observed in cardiac adaptations to cold.

Female rodents have more active BAT than males [38,39]. Some evidence shows that this could be the case in women, too [40,41]. Women have an increased number of brown adipocytes compared to men [41]. Estradiol may be responsible for this. Pedersen and collaborators showed that treatment of ovariectomized rats with β-estradiol prevented the reduction in BAT *Ucp1* mRNA expression induced by ovariectomy [42].

Housing mice at thermoneutrality provides an opportunity to assess the degree of stress that ambient housing temperatures represent for mice. Globally, for control mice, the differences between mice at 22 °C and 30 °C were relatively mild. Except for food consumption in mice housed at 22 °C, we also observed that females had less lean mass and had an enlarged left atrium. Only food consumption was increased at ambient temperature in males.

Differences became evident when the mice were subjected to the MHS. Thermoneutral groups had a blunted hypertrophic response compared to those housed at 22 °C, and left atrial enlargement was also reduced. The MHS also modulated fewer diastolic echocardiography parameters. Interestingly, these potential benefits of thermoneutrality were not observed in the expression of gene markers for hypertrophy and extracellular matrix remodelling. For instance, collagens were more expressed in the LV of MHS mice at 30 °C compared to those at 22 °C. *Nppa* and *Postn* were also more expressed in MHS males at thermoneutrality.

Cardiac adaptations during the acclimatization period of mice transiting from ambient room temperature to 30 °C may have coincided with those induced by the MHS stress, since only one week separated the two. However, thermoneutrality helped reduce the hypertrophic response to MHS. Substantial morphological and functional changes in cardiac physiology are not induced by the cold stress of being housed at 22 °C, but it may fragilize the mice exposed to the MHS.

Cold exposure creates a demand for increased thermogenesis. Brown adipose tissue is an essential mammal heat producer activated by the β3-adrenergic receptor (β3-AR) in mice. Uncoupled protein 1 (UCP1) plays a central role in this response to cold. Cold stimuli are transmitted to the hypothalamus through the skin, indirectly releasing norepinephrine to activate the cAMP-protein kinase A (PKA) signalling pathway via the β3-AR. PKA phosphorylation activates factors that lead to the activation of PGC-1α expression. This activates intranuclear UCP1 production, which then migrates to the mitochondria to produce heat [43].

In addition to its role in thermogenesis, BAT has been shown to produce factors (batokines) that can protect the cardiovascular system, particularly the heart, in pathological situations. Evidence for the modulation of cardiovascular function exists in the context of pathological states, such as hypertension, atherosclerosis, and ischemia/reperfusion injury, for several of these batokines (FGF21, neuregulin 4, 12,13-diHOME, and BAT-derived microRNAs) [11]. Activation of the β3-AR in BAT has been shown to exhibit cardioprotective effects by suppressing exosomal inducible nitric oxide synthase (iNOS) originating from the BAT [44].

As we observed here, the BAT in mice at thermoneutrality presents histological similarities with white adipose tissue, whereas browning is evident at 22 °C and 10 °C. This is accompanied by increased UCP1 levels, which become more elevated as the cold stress intensifies. The activation of the BAT by cold exposure may help produce cardioprotective factors or inhibit others, such as iNOS [11,45]. These putative benefits of BAT activation were entirely superseded by the heart’s obligatory response to increase its cardiac output in a cold environment. Those factors did not protect the heart against the MTS in the cold-stressed groups, although they may have helped limit the adverse response to low temperatures. It would be interesting to investigate the action of pharmaceutical agonists or antagonists of β3-AR to modulate this signalling pathway and test if BAT activation without a cold stress could benefit the response to MHS. For instance, Lin and colleagues observed that mirabegron, a β3-AR agonist, could activate the BAT and inhibit the release of exosomal iNOS in the angiotensin II (Ang II)-induced mouse cardiac hypertrophy model [45].

Food consumption was reduced in mice housed at 30 °C, resulting in fewer calories from fat being consumed. In our model, the HFD does not cause cardiac hypertrophy in males, but does in females. Only in mice housed at 22 °C did we observe an increase in calories consumed in mice fed the HFD. As previously described by Chen and collaborators [46], housing mice at thermoneutrality reduces blood pressure (BP), but this reduction could be reversed by feeding mice with an HFD. They also test a pressure overload stress in their mice by performing a transverse aortic constriction (TAC). A significant increase in indexed heart weight was observed in TAC mice at 22 °C. CH tended to be lower in the thermoneutral mice.

AngII is the primary hypertrophic stress in our model [13], and its effects were almost completely blunted in mice under thermoneutral conditions. We did not record BP in this study. We previously observed that AngII infusion alone significantly increased BP in males, not in females [13]. The cardiac hypertrophy induced by AngII was equivalent between the sexes, regardless of BP. BP was likely lower in the MHS mice at 30 °C, although feeding with the HFD could have partially counteracted the benefits of thermoneutrality on BP. Since baseline cardiac work is likely lower in mice at thermoneutrality, they may possess better reserves and be more resilient when a pathological stress is added.

This needs to be explored further, but apart from cardiac hypertrophy, the benefits of housing at thermoneutrality compared to standard room temperature were relatively mild in our mice. Does this mean that the cold stress caused by housing at standard room temperature should not be taken into account in preclinical studies in the field of cardiology? Our results suggest that a more stringent pathological stress should be applied to mice at thermoneutrality to achieve a similar cardiac response as observed in mice at 22 °C. Studying animals at 30 °C is also impractical for most researchers, as housing facilities were not designed to accommodate large numbers of animals at this temperature.

Other factors, such as sedentary behaviour, housing mice alone in their cage or a group, environmental enrichment, and other extrinsic factors, also influence the animal’s response to stress [47,48]. This emphasizes the need for standardization in housing conditions to increase reproducibility of research results obtained from mice.

The MHS induced browning of the BAT in mice at thermoneutrality, accompanied by elevated *Ucp1* gene expression and decreased adipocyte size, suggesting the activation of this fat depot. The MHS combines two factors with potential opposite effects on general adiposity. We previously observed that the HFD for four weeks increased body weight in mice compared to a standard diet, and AngII did the opposite in males. Females’ body weight remained unchanged after AngII. Combining the two, as for the MHS, resulted in similar body weights after four weeks in animals at 22 °C [13].

The control of adiposity by the RAAS is believed to be mediated by the Angiotensin II receptor type II (AT2R). The knock-out of this receptor results in increased obesity in mice fed an HFD, and its activation with a specific agonist (C21) blunts weight gain [49,50,51].

C21 can also lead to BAT activation (upregulation of *Ucp1*) and the browning of white adipose tissue in mice [51]. In our animals at thermoneutrality, it seems that the effects of AngII on the BAT were preponderant. It is unclear how it altered batokines production in our mice, which may have contributed to the reduction in CH in the MHS mice.

Cold exposure has been identified as a risk factor for the development of various human diseases, including cardiovascular diseases [52]. Previous studies have reported an association (positive and negative). A recent meta-analysis on the subject has concluded that each 1 °C decrease below the reference point was associated with an increase in the prevalence of cardiovascular disease between 1 and 1.5%. Our results may offer some insights into the higher prevalence of heart disease in colder regions of the Earth.

Still, comparisons between humans and mice, in the context of the study, are difficult to make. The capacity for thermogenesis in humans is limited compared to that of most mammals; accordingly, strategies for coping with cold are different. In terms of evolution, the presence of humans in colder regions of the Earth is relatively short compared to animals that had time to adapt to these conditions.

## 5. Study Limitations

This study used a two-hit model to induce HFpEF that resembles human HFpEF. It is still not representative of the entire spectrum of HFpEF patients. In addition to being housed in a cold-stressed environment, being housed alone may have changed the mice’s social environment.

The translation of our results to humans is challenging, but they emphasize that mouse housing temperatures are a factor in the development and study of preclinical animal models.

Several parameters were not measured in this study, including BP, serum catecholamines, animal metabolic activity, or daily activity, a more complete histological characterization of the myocardium, which may have helped better understand our observations.

A more detailed characterization of myocardial immune cells could enhance our understanding of the role of inflammation in cardiac adaptations to cold.

## 6. Conclusions

Our results indicate that housing temperatures influence the cardiac response to a stressor that induces HFpEF. In itself, housing at 10 °C demands essential cardiovascular adaptations to increase the cardiac output. Although those adaptations may be, in large part, reversible once the animal is returned to warmer housing environments, suggesting physiological cardiac remodelling, it likely makes the heart more fragile to respond to an additional stress, the MHS in this case.

On the other hand, thermoneutrality blunts cardiac hypertrophy without reducing myocardial fibrosis. This suggests a more complex interplay between factors known to be reduced in these conditions (e.g., catecholamines, BP, and others) on cardiac remodelling in a pathological stress.

More studies are needed to better understand the relative contribution of these two stresses, cold and MHS, to the complex array of mechanisms underlying the cardiac adaptations observed in mice.

## Figures and Tables

**Figure 1 biomedicines-13-01900-f001:**
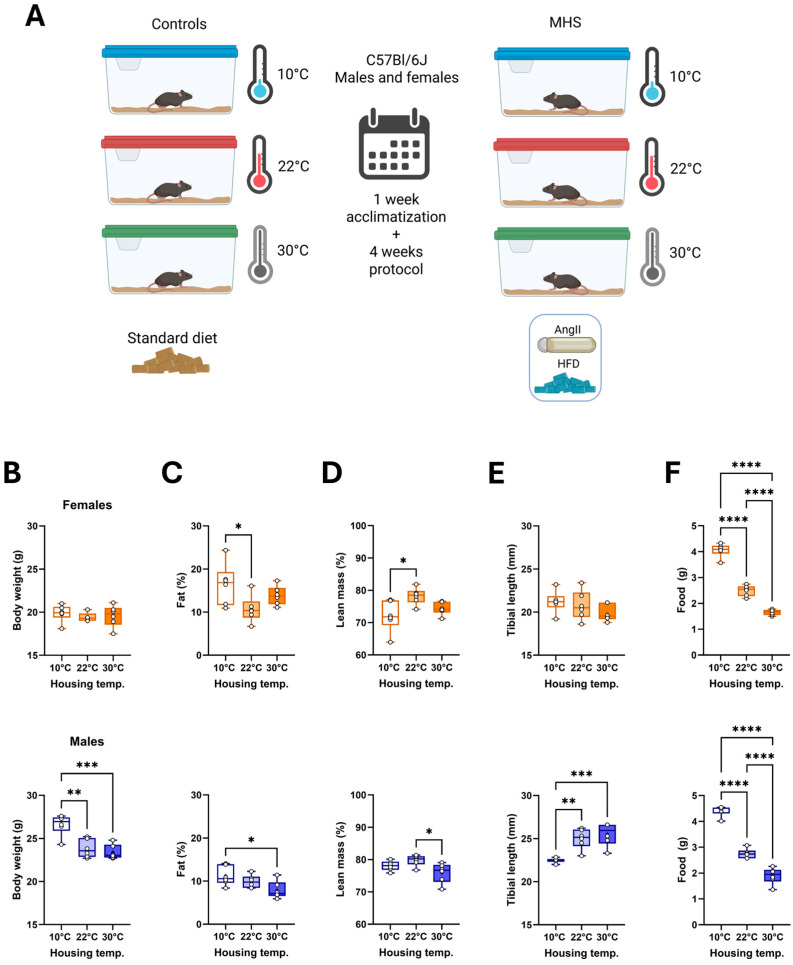
Effects of three different housing temperatures (10 °C, 22 °C, or 30 °C) on mouse body composition and food consumption. (**A**) Schematic representation of the experimental design (Created in BioRender. Couet, J. (https://BioRender.com/r27xz47, 16 June 2025)). Eight-week-old C57BL/6J male and female mice were housed individually for 5 weeks (1 week of acclimatization and 4 weeks for the protocol). All mice were fed a standard diet described in the Materials and Methods section. After one week, the MHS was started in half of the mice for four weeks. (**B**) Body weight, (**C**) Body fat (%), (**D**) Lean mass (%), (**E**) Tibial length, and (**F**) Daily food consumption. Graphs of females (top, orange) and males (bottom, blue). Results are expressed as mean ± standard error of the mean (SEM). One-way ANOVA followed by Holm–Sidak post-test. *: *p* < 0.05, **: *p* < 0.01, ***: *p* < 0.001, and ****: *p* < 0.0001 between indicated groups (n = 6 mice/group).

**Figure 2 biomedicines-13-01900-f002:**
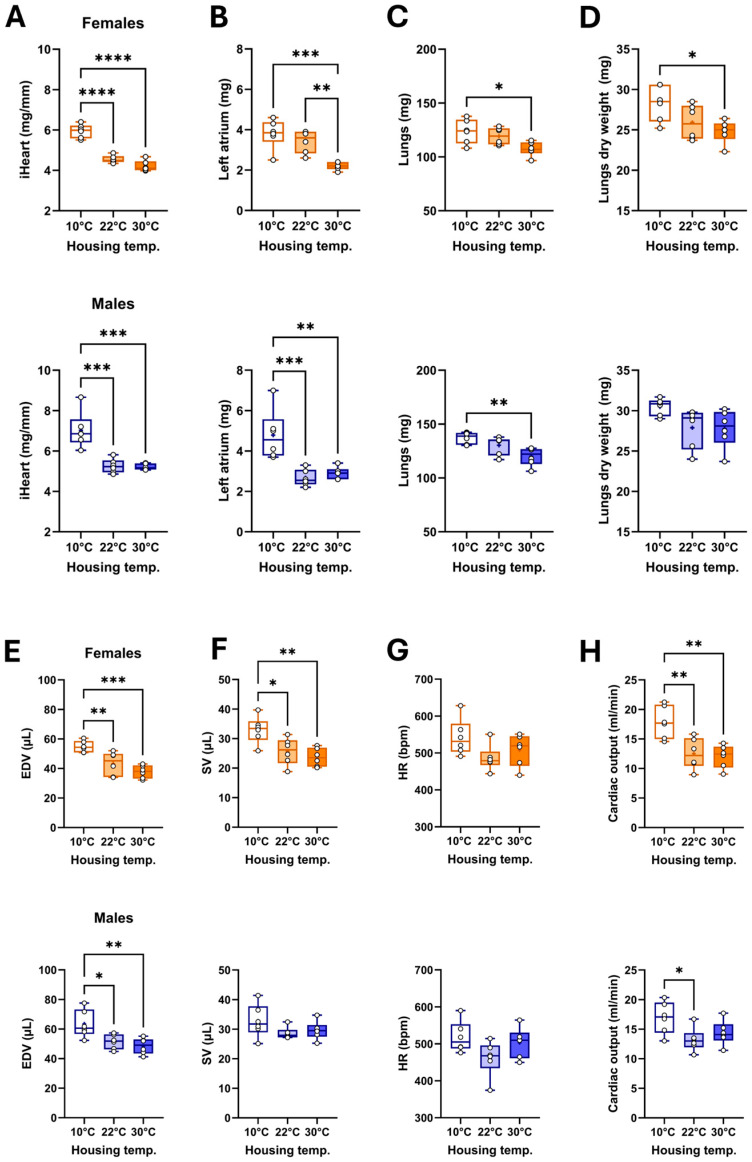
Effects of three different housing temperatures (10 °C, 22 °C, or 30 °C) on mouse cardiac morphology and function. (**A**) Indexed heart weight for the tibial length, (**B**) Left atrial weight, (**C**) Lungs wet weight, (**D**) Lungs dry weight, (**E**) End-diastolic volume (EDV), (**F**) LV stroke volume (SV), (**G**) Heart rate (HR), and (**H**) Cardiac output. Results are expressed as mean ± SEM. One-way ANOVA followed by Holm–Sidak post-test. *: *p* < 0.05, **: *p* < 0.01, ***: *p* < 0.001, and ****: *p* < 0.0001 between indicated groups (n = 6 mice/group).

**Figure 3 biomedicines-13-01900-f003:**
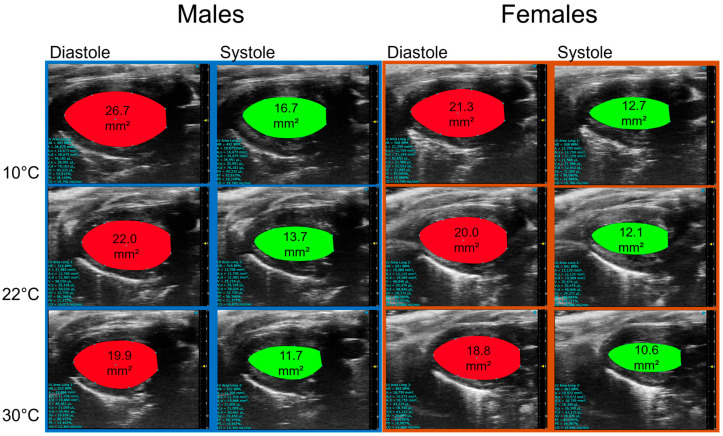
Representative B-mode LV diastolic and systolic tracings of mice 5 weeks after being housed at 10 °C, 22 °C, or 30 °C.

**Figure 4 biomedicines-13-01900-f004:**
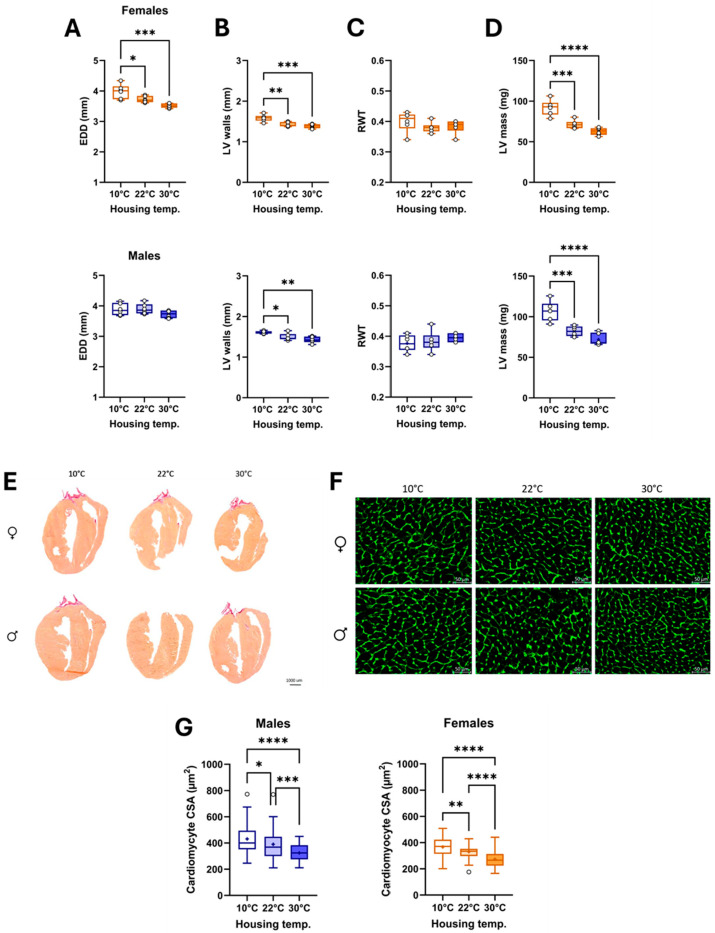
Effects of three different housing temperatures (10 °C, 22 °C, or 30 °C) on LV dimensions and cardiomyocyte cross-sectional area (CSA). (**A**) End-diastolic diameter (EDD), (**B**) LV walls (posterior + septal) thickness, (**C**) Relative wall thickness (RWT), (**D**) LV mass. (**E**) Representative images of picrosirius red staining of female and male heart sections, (**F**) Representative images of WGA-FITC staining from LV sections of the various indicated groups. (**G**) CSA of cardiomyocytes was quantified by WGA-FITC staining. Results are expressed as mean ± SEM. One-way ANOVA followed by Holm–Sidak post-test. *: *p* < 0.05, **: *p* < 0.01, ***: *p* < 0.001, and ****: *p* < 0.0001 between indicated groups (n = 6 mice/group).

**Figure 5 biomedicines-13-01900-f005:**
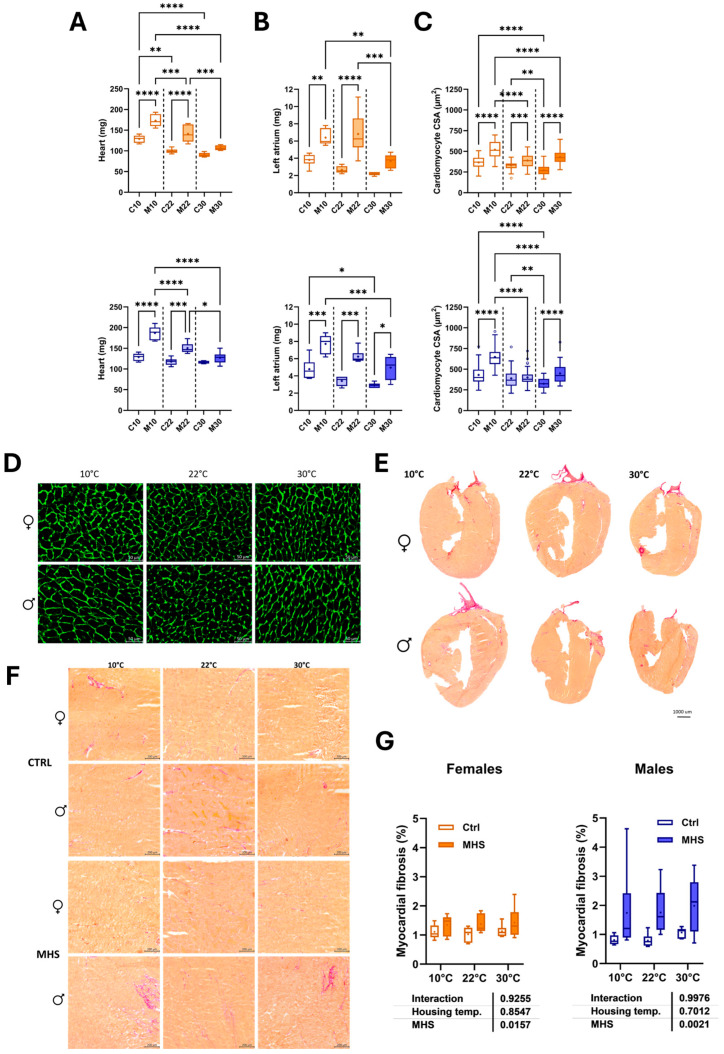
Effects of MHS (M) on mice housed at three different temperatures (10 °C, 22 °C, or 30 °C) on cardiac morphology and cardiomyocyte area compared to controls (**C**). (**A**) Heart weight, (**B**) Left atrial weight, (**C**) Cross-sectional area, (**D**) Cross-sectional area of cardiomyocytes stained with WGA-FITC, (**E**) Representative images of picrosirius red staining of MHS female and male heart sections, (**F**) Magnification of a mid-posterior wall section of control and MHS LV sections. Bar scale: 200 µm. (**G**). Myocardial fibrosis (picrosirius red staining). Results are expressed as mean ± SEM. Two-way ANOVA followed by Holm–Sidak post-test. *: *p* < 0.05, **: *p* < 0.01, ***: *p* < 0.001, and ****: *p* < 0.0001 between indicated groups (n = 6 mice/group).

**Figure 6 biomedicines-13-01900-f006:**
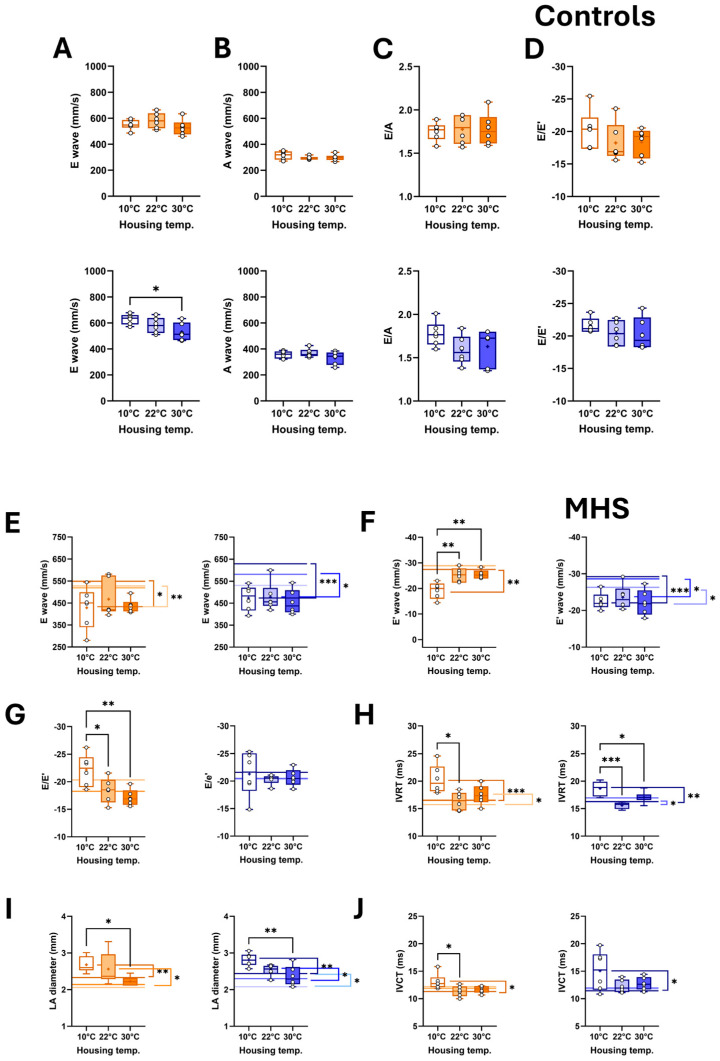
Effects of MHS (M) on diastolic function echocardiography parameter in mice housed at three different temperatures (10 °C, 22 °C, or 30 °C). Controls: (**A**) E wave velocity, (**B**) A wave velocity, (**C**) E/A ratio and (**D**) E/E’ ratio. MHS: (**E**) E wave velocity, (**F**) E’ wave velocity, (**G**) E/E’ ratio, (**H**) Isovolumetric relaxation time (IVRT), (**I**) LA diameter, and (**J**) Isovolumetric contraction time (IVCT). Lines in Graphs E to J represent the average value of this parameter in control mice. Darker to lighter colours represent 10°, 22°, and 30 °C, respectively. Results are expressed as mean ± SEM—one-way ANOVA followed by Holm–Sidak post-test. Comparisons to controls were analyzed using T-tests. *: *p* < 0.05, **: *p* < 0.01, ***: *p* < 0.001, and ****: *p* < 0.0001 between indicated groups (n = 6 mice/group).

**Figure 7 biomedicines-13-01900-f007:**
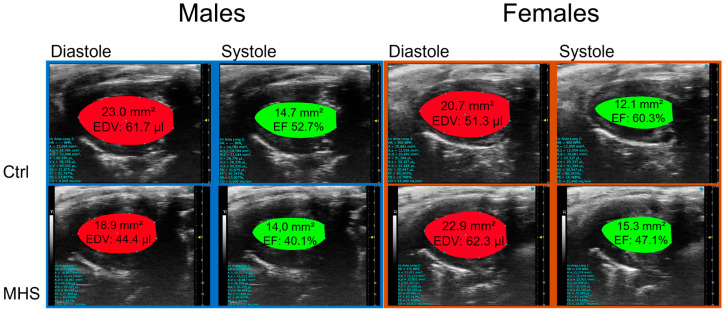
Representative B-mode LV diastolic and systolic tracings of control (Ctrl) and MHS housed at 10 °C.

**Figure 8 biomedicines-13-01900-f008:**
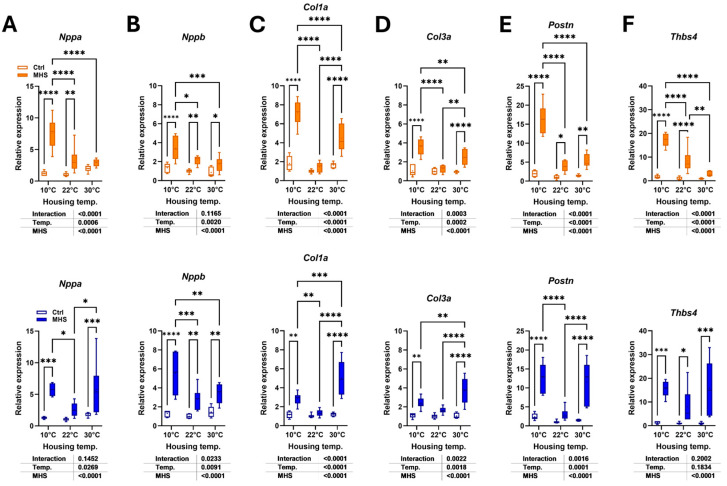
Modulation of LV gene expression after MHS in mice housed at three different temperatures. (**A**) *Nppa*, atrial natriuretic peptide. (**B**) *Nppb*, brain natriuretic peptide. (**C**) *Col1a1*, Collagen 1α1. (**D**) *Col3a1*, Collagen 3α1. (**E**) *Postn*, periostin. (**F**) *Tbsp4*, thrombospondin 4. Data are represented as mean ± SEM (n = 6 per group). Two-way ANOVA followed by Holm–Sidak post-test. *: *p* < 0.05, **: *p* < 0.01, ***: *p* < 0.001, and ****: *p* < 0.0001 between indicated groups.

**Figure 9 biomedicines-13-01900-f009:**
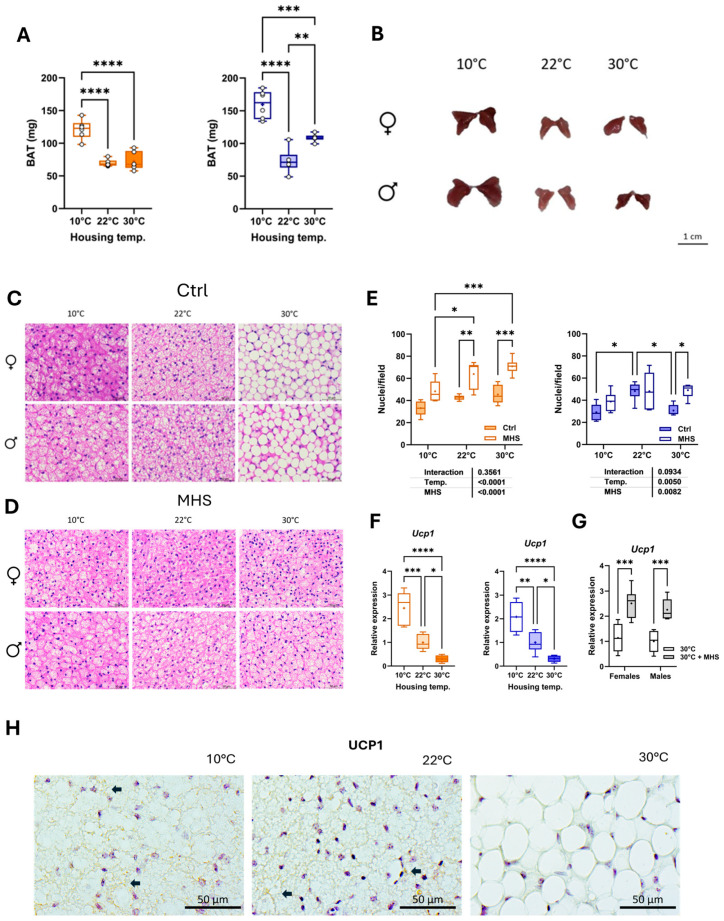
Effects of housing temperature and MHS on brown adipose morphology. (**A**) Scapular brown fat depot weight (BAT). (**B**) Representative pictures of BAT depots for each control group. (**C**) Hematoxylin/eosin staining of the BAT section from a mouse in each control group. (**D**) Hematoxylin/eosin staining of a BAT section from a mouse of each MHS group. (**E**) Number of nuclei/field. (**F**) *Ucp1* or Uncoupled protein 1 gene expression in the BAT of control mice. (**G**) BAT *Ucp1* expression in mice at 30 °C after the MHS. (**H**) BAT sections from a male control mouse for each group labelled with a UCP1 antibody. Arrows point to positive labelling. None was detected in BAT at 30 °C. Data are represented as mean ± SEM (n = 6 per group). One- or two-way ANOVA followed by Holm–Sidak post-test. *: *p* < 0.05, **: *p* < 0.01, ***: *p* < 0.001, and ****: *p* < 0.0001 between indicated groups.

**Figure 10 biomedicines-13-01900-f010:**
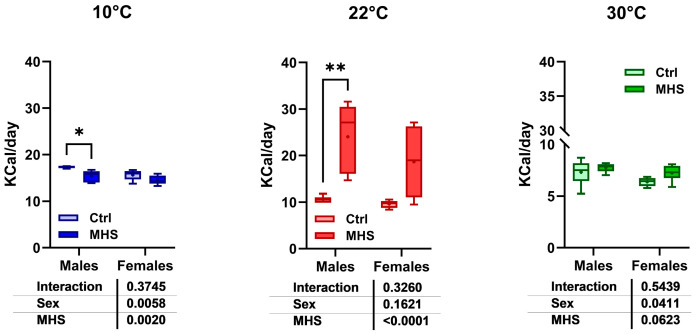
Average daily food consumption (KCalories per day; Kcal/day) for each mouse group. Data are represented as mean ± SEM (n = 6 per group). Two-way ANOVA followed by Holm–Sidak post-test. *: *p* < 0.05, **: *p* < 0.01 between indicated groups.

**Table 1 biomedicines-13-01900-t001:** Sequences of primers used in this study.

Symbol	Description	Forward Sequence Reverse Sequence
Col1a1	Collagen Type I Alpha 1 Chain	5′-CAT TGT GTA TGC AGC TGA CTT C-3′ 5′CGC AAA CAC TCT ACA TGT CTA GG-3′
Col3a1	Collagen Type III Alpha 1 Chain	5′-TCT CTA GAC TCA TAG GAC TGA CC-3′ 5′ TTC TTC TCA CCC TTC TTC ATC C-3′
Nppa	Natriuretic Peptide B	5′-CTC CTT GGC TGT TAT CTT CGG-3′ 5′-GGG TAG GAT TGA CAG GAT TGG-3′
Nppb	Natriuretic Peptide A	5′-AGG TGA CAC ATA TCT CAA GCT G-3′ 5′-CTT CCT ACA ACA TCA GTG C-3′
Ppia	Cyclophilin A	5′-TTC ACC TTC CCA AAG ACC AC-3′ 5′-CAA ACA CAA ACG GTT CCC AG-3′
Postn	Periostin	5′-GCT TTC GAG AAA CTG CCA CG-3′ 5′-ATG GTC TCA AAC ACG GCT CC-3′
Thbs4	Thrombospondin 4	5′-GAT ACT GAC GGG GAT GGG AG-3′ 5′-CGT CAC TGT CTT GGT TGG TG-3′
Ucp1	Uncoupled protein 1	5′-GCT TCT ACG ACT CAG TCC AA-3′ 5′-CTC TGG GCT TGC ATT CTG AC-3′

**Table 2 biomedicines-13-01900-t002:** Echo data in male and female mice after MHS (M) at three different housing temperatures (10 °C, 22 °C, and 30 °C). Controls (C). Standard echo left ventricle parameters after 4 weeks of MHS and after 4 weeks. Control mice were studied in parallel Echo exams as described in the Methods section.

Males						
Parameters	C10 (n = 6)	M10 (n = 6)	C22 (n = 6)	M22 (n = 6)	C30 (n = 6)	M30 (n = 6)
PWd, mm	0.85 ± 0.012	1.11 ± 0.035 **^d^**	0.78 ± 0.017	1.05 ± 0.032 **^d^**	0.76 ± 0.011	1.10 ± 0.038 **^d^**
IVSd, mm	0.76 ± 0.010	0.93 ± 0.024 **^d^**	0.71 ± 0.017	0.94 ± 0.043 **^c^**	0.67 ± 0.019	0.89 ± 0.027 **^d^**
EDD, mm	4.29 ± 0.101	4.12 ± 0.142	3.90 ± 0.057	3.54 ± 0.092 **^b^**	3.73 ± 0.045	3.22 ± 0.091 **^c^**
ESD, mm	3.25 ± 0.104	3.18 ± 0.128	2.93 ± 0.088	2.42 ± 0.116 **^b^**	2.57 ± 0.051	2.09 ± 0.102 **^b^**
RWT	0.38 ± 0.009	0.50 ± 0.031 **^b^**	0.38 ± 0.012	0.57 ± 0.032 **^c^**	0.38 ± 0.008	0.62 ± 0.023 **^d^**
LV mass, mg	133 ± 5.2	171 ± 3.8 **^d^**	103 ± 2.6	132 ± 5.0 ^**c**^	90 ± 3.1	115 ± 8.7 **^a^**
EDV, µL	63 ± 3.2	53 ± 2.6 **^a^**	51 ± 1.7	39 ± 2.5 **^b^**	49 ± 1.9	37 ± 2.5 **^b^**
ESV, µL	31 ± 1.9	29 ± 1.6	23 ± 1.4	17 ± 2.0 **^b^**	19 ± 1.3	14 ± 1.7 **^a^**
SV, mm	32 ± 1.9	24 ± 1.8 **^b^**	29 ± 0.7	22 ± 1.3 **^c^**	30 ± 1.1	23 ± 1.1 **^b^**
HR, bpm	518 ± 14.5	579 ± 13.0	461 ± 16.4	495 ± 4.5	503 ± 14.1	532 ± 13.3
EF, %	52 ± 1.8	44 ± 1.8 **^a^**	56 ± 1.6	58 ± 1.1	62 ± 1.6	63 ± 1.4
CO, mL/min	16.9 ± 0.93	13.7 ± 0.94 **^a^**	13.9 ± 0.69	9.8 ± 0.75 **^a^**	14.4 ± 0.71	12.4 ± 0.54
**Females**						
**Parameters**	**C10 (n = 6)**	**M10 (n = 6)**	**C22(n = 6)**	**M22(n = 6)**	**C30(n = 6)**	**M30(n = 6)**
PWd, mm	0.85 ± 0.017	1.11 ± 0.043 **^c^**	0.72 ± 0.009	0.96 ± 0.034 **^d^**	0.72 ± 0.009	0.89 ± 0.018 **^d^**
IVSd, mm	0.73 ± 0.097	0.96 ± 0.037 **^c^**	0.70 ± 0.011	0.87 ± 0.034 **^c^**	0.67 ± 0.011	0.88 ± 0.030 **^d^**
EDD, mm	3.98 ± 0.020	3.89 ± 0.067	3.73 ± 0.037	3.75 ± 0.148	3.51 ± 0.025	3.27 ± 0.050 **^b^**
ESD, mm	2.88 ± 0.098	3.02 ± 0.090	2.67 ± 0.054	2.67 ± 0.054	2.30 ± 0.030	2.15 ± 0.044 **^b^**
RWT	0.40 ± 0.014	0.53 ± 0.024 **^c^**	0.38 ± 0.006	0.50 ± 0.034	0.40 ± 0.004	0.54 ± 0.015 **^d^**
LV mass, mg	115 ± 4.8	161 ± 8.8 **^b^**	89 ± 2.2	127 ± 3.9 **^d^**	78 ± 2.0	98 ± 4.1 **^b^**
EDV, µL	55 ± 1.6	52 ± 2.4	43 ± 2.7	50 ± 3.3	38 ± 1.5	34 ± 2.25
ESV, µL	22 ± 0.8	27 ± 1.1 **^b^**	18 ± 1.3	24 ± 2.0 **^a^**	14 ± 0.8	12 ± 1.3
SV, mm	33 ± 1.9	26 ± 1.9 **^a^**	26 ± 1.6	26 ± 1.8	24 ± 1.1	21 ± 1.1
HR, bpm	542 ± 20.1	531 ± 15.3	486 ± 12.1	515 ± 11.5	508 ± 14.8	513 ± 8.4
EF, %	60 ± 2.0	49 ± 3.2 **^a^**	59 ± 1.3	59 ± 1.1	63 ± 1.4	64 ± 2.0
CO, mL/min	17.8 ± 1.12	13.7 ± 1.09 **^a^**	12.5 ± 0.90	13.6 ± 0.96	12.0 ± 0.67	11.0 ± 0.66

PWd: diastolic posterior wall thickness, IVSd: diastolic interventricular septum, EDD: end-diastolic LV diameter, ESD: end-systolic LV diameter, RWT: relative wall thickness, EDV: end-diastolic volume, ESV: end-systolic volume, SV: stroke volume, HR: heart rate, EF: ejection fraction, CO: cardiac output. Results are expressed as the mean ± SEM. P values were calculated using Student’s T-test. a: *p* < 0.05 vs. controls, b: *p* < 0.01, c: *p* < 0.001, and d: *p* < 0.0001.

## Data Availability

The authors will make the raw data supporting the conclusions of this article available on request.

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
