# Peer review of "Cold Exposure Exacerbates Cardiac Dysfunction in a Model of Heart Failure with Preserved Ejection Fraction in Male and Female C57Bl/6J Mice"

_biomedicines, 2025, doi:10.3390/biomedicines13081900_

Round 1
Reviewer 1 Report
Comments and Suggestions for Authors
The problem of cardiovascular diseases in cold-climate conditions is an acute problem. In the northern countries separate laboratories are being created to study the mechanisms by which the heart adapts to cold climates.
This work is excellently performed. The murine model used corresponds to the stated goal and objectives of the study. The study design (groups of mice housed under three temperature regimes with HFpEF, as well as control groups under similar conditions) allow us to form a complete picture of changes in the cardiovascular system (and myocardium in particular) to low temperature.
Particularly impressive is the thoroughness with which the structural and functional parameters of myocardium were assessed by EchoCG in different groups, as well as the morphological study of myocardium and assessment of LV gene expression.
While reading the manuscript, a few points arose that I would like to clarify.
- During the morphological examination of the myocardium, were only fibrotic changes evaluated? Were inflammatory changes in the myocardium noted? For example, was lymphocytic infiltration due to reactive inflammation caused by oxidative stress noted?
- Any animal study provides a basis for drawing analogies to the human body. Perhaps it makes sense to devote at least one paragraph in the discussion to the prospects of practical application of the findings in clinical practice. For example, include data from observational studies (e.g. a systematic review and meta-analysis of cold exposure and cardiovascular disease outcomes. Front Cardiovasc Med. 2023 Mar 27;10:1084611. doi:10.3389/fcvm.2023.1084611) and within the context of the findings to discuss the potential for mitigating the adverse effects of a cold climate on the cardiovascular system.
Author Response
- During the morphological examination of the myocardium, were only fibrotic changes evaluated? Were inflammatory changes in the myocardium noted? For example, was lymphocytic infiltration due to reactive inflammation caused by oxidative stress noted?
This is an interesting comment. We only evaluated fibrotic changes in the myocardium. We never assessed the lymphocytic infiltration in our model. We looked at macrophages in the past but did not notice important changes in our model. We will include this as a perspective in the discussion section.
- Any animal study provides a basis for drawing analogies to the human body. Perhaps it makes sense to devote at least one paragraph in the discussion to the prospects of practical application of the findings in clinical practice. For example, include data from observational studies (e.g. a systematic review and meta-analysis of cold exposure and cardiovascular disease outcomes. Front Cardiovasc Med. 2023 Mar 27;10:1084611. doi:10.3389/fcvm.2023.1084611) and within the context of the findings to discuss the potential for mitigating the adverse effects of a cold climate on the cardiovascular system.
We added a few sentences discussing the potential translational value in human of our observations.
Reviewer 2 Report
Comments and Suggestions for Authors
The results of this work further indicate that experiments performed on mice reared at 22OC (and usually such a temperature is maintained in the vivarium) may not be conclusive due to heat stress. How much this finding should provoke us to additional research and analysis of the results on metabolic syndrome, obesity and cardiovascular disease?
I would like to ask the authors for a few more sentences explaining the differences in thermoregulatory capacity in humans and mice. How does cold adaptation differ in these species, in the context of the parallel development of HFpEF and metabolic syndrome in women and female mice?
There is an error in the text - line 162 should quote table 2, not 1. Table 2 is poorly readable. I think the p-value (a, b, c, and d) in this table could be shown in bold.
Author Response
The results of this work further indicate that experiments performed on mice reared at 22OC (and usually such a temperature is maintained in the vivarium) may not be conclusive due to heat stress. How much this finding should provoke us to additional research and analysis of the results on metabolic syndrome, obesity and cardiovascular disease?
I would like to ask the authors for a few more sentences explaining the differences in thermoregulatory capacity in humans and mice. How does cold adaptation differ in these species, in the context of the parallel development of HFpEF and metabolic syndrome in women and female mice?
We added a few sentences discussing differences between mice and humans related to cold exposure and about the potential translational value in human of our observations.
There is an error in the text - line 162 should quote table 2, not 1. Table 2 is poorly readable. I think the p-value (a, b, c, and d) in this table could be shown in bold.
We made the change.
Reviewer 3 Report
Comments and Suggestions for Authors
This is a well-structured and timely study investigating the effects of housing temperature on cardiac remodeling and function in a murine model of HFpEF induced by Ang II and high-fat diet (MHS). The authors have addressed an important, yet often overlooked, experimental variable—ambient temperature—in cardiovascular research, and their findings have clear implications for the design and interpretation of preclinical HFpEF models. The use of both male and female animals is a notable strength.
However, several key methodological and interpretive issues need to be addressed before the manuscript can be considered for publication. In its current form, the study lacks essential physiological data, and some conclusions are speculative without mechanistic support.
ï‚· Given that Ang II is a hypertensive agent and that both cold exposure and high-fat diet influence hemodynamics, it is a significant omission that blood pressure (BP) was not measured. Without BP data, it is difficult to interpret the contribution of hemodynamic load to the observed hypertrophy and functional changes. If BP data are available, they should be included. If not, this limitation must be clearly acknowledged and discussed in the manuscript.
ï‚· The study suggests a cardioprotective or modulatory role of BAT via “batokines,” yet these factors (e.g., FGF21, 12,13-diHOME, neuregulin 4) were not measured. UCP1 mRNA was assessed, but no protein-level confirmation or systemic marker of BAT activation was presented. Either include such measurements or revise the discussion to avoid overinterpretation of the BAT–heart interaction.
ï‚· Although echocardiographic indices (e.g., E/E′, IVRT) are suggestive of diastolic dysfunction, there is no assessment of myocardial stiffness, fibrosis quality (e.g., collagen I vs III), or titin isoform expression. The authors should moderate their interpretation or add further molecular data if available.
ï‚· The manuscript mentions some sex-specific effects (e.g., more pronounced dysfunction in females), but these findings are not systematically analyzed. Provide a clearer breakdown of sex differences in cardiac outcomes, and discuss potential mechanisms (e.g., hormonal regulation, estrogen signaling) if possible.
ï‚· The one-week acclimatization period before starting the MHS protocol may not be sufficient for full thermal adaptation, particularly at 30°C. The authors should discuss how this overlap might have influenced their findings.
ï‚· While CSA and Picrosirius Red staining were used, the assessment of myocardial inflammation, capillary density, or vascular remodeling is absent. Additional histological markers (e.g., CD68, CD31) would strengthen the tissue-level interpretation, or the authors should address this as a limitation.
ï‚· Some statements about cold stress “fragilizing” the heart or the protective role of thermoneutrality are presented without sufficient mechanistic evidence. The discussion should be more balanced, highlighting hypotheses rather than definitive claims.
ï‚· Please ensure that all figures are adequately labeled with group sizes and statistical annotations.
ï‚· Consider adding plasma catecholamine measurements, if available, to support claims of sympathetic activation.
ï‚· Clarify whether housing temperature also affected heart rate variability or arrhythmogenesis, which could be relevant to HFpEF physiology.
Author Response
- Given that Ang II is a hypertensive agent and that both cold exposure and high-fat diet influence hemodynamics, it is a significant omission that blood pressure (BP) was not measured. Without BP data, it is difficult to interpret the contribution of hemodynamic load to the observed hypertrophy and functional changes. If BP data are available, they should be included. If not, this limitation must be clearly acknowledged and discussed in the manuscript.
The reviewer is right and this is missing. We already measured BP in previous studies and showed that the combination of AngII and the HFD increased BP in our model in mice at room temperature (mean bp raised by a little less than 20mmHg in males). The increase was more marked in males than in females although cardiac hypertrophy was similar in both sexes. We believed that direct hypertrophic effects of AngII on the heart are probably more important than the raise of the afterload from hypertensin, which is relatively mild.
- The study suggests a cardioprotective or modulatory role of BAT via “batokines,” yet these factors (e.g., FGF21, 12,13-diHOME, neuregulin 4) were not measured. UCP1 mRNA was assessed, but no protein-level confirmation or systemic marker of BAT activation was presented. Either include such measurements or revise the discussion to avoid overinterpretation of the BAT–heart interaction.
We included UCP1 IHC in Figure 9 showing labeling at 10 and 22C.
- Although echocardiographic indices (e.g., E/E′, IVRT) are suggestive of diastolic dysfunction, there is no assessment of myocardial stiffness, fibrosis quality (e.g., collagen I vs III), or titin isoform expression. The authors should moderate their interpretation or add further molecular data if available.
Diastolic dysfunction is most often evaluated using echo parameters in mice (as in humans) without characterizing the myocardium. Other signs point towards DD including LA enlargement (diameter and weight), E wave reduction, stroke volume reduction without loss of ejection fraction indicating LV filling anomalies. Myocardial stiffness is obviously one factor involved in the development of diastolic dysfunction, the other is atrial myopathy. We think that diastolic dysfunction is present in the animals, which would be expected in a model leading to LV concentric remodelling (de facto increased stiffness since walls are thicker than normal) and LA enlargement. What it the relative contribution of the LV and LA to the diastolic dysfunction is unknown but anomalies are present.
- The manuscript mentions some sex-specific effects (e.g., more pronounced dysfunction in females), but these findings are not systematically analyzed. Provide a clearer breakdown of sex differences in cardiac outcomes, and discuss potential mechanisms (e.g., hormonal regulation, estrogen signaling) if possible.
We added a comment on this in the discussion. Females are usually more prone to develop diastolic dysfunction than males, which tends to develop systolic dysfunction. Interestingly, at 10C, both sexes lost systolic dysfunction after the MHS stress.
- The one-week acclimatization period before starting the MHS protocol may not be sufficient for full thermal adaptation, particularly at 30°C. The authors should discuss how this overlap might have influenced their findings.
This is a valid point. It is possible that acclimatization may have been too short before starting the MHS protocol. All groups were treated the same,though. Total cold or warm exposures were equivalent in the end and it is clear that animals at 30C faired better in the end.
- While CSA and Picrosirius Red staining were used, the assessment of myocardial inflammation, capillary density, or vascular remodeling is absent. Additional histological markers (e.g., CD68, CD31) would strengthen the tissue-level interpretation, or the authors should address this as a limitation.
We added this as a limitation.
- Some statements about cold stress “fragilizing” the heart or the protective role of thermoneutrality are presented without sufficient mechanistic evidence. The discussion should be more balanced, highlighting hypotheses rather than definitive claims.
We modified the discussion to take into account the comments of the Reviewer. Still, housing at 10°C indeed fragilized mice to the MHS compared to mice housed at higher temperatures.
- Please ensure that all figures are adequately labeled with group sizes and statistical annotations.
We made changes in the table and put in bold the letters highlighting statistical differences. Group sizes and statistical annotations are already indicated in the legend of all figures.
- Consider adding plasma catecholamine measurements, if available, to support claims of sympathetic activation.
We did not measure catecholamines in our animals although numerous evidence for this about cold exposure exist in the literature. We add a comment about this in the limitations
- Clarify whether housing temperature also affected heart rate variability or arrhythmogenesis, which could be relevant to HFpEF physiology.
We did not measure this. We added a mention about this in the discussion. Most likely HR variability is less in the MHS animals. We never noticed blatant arrhythmia in the EKG we record during echo, though. There never recorded A fib although larger LA is a important risk factor for A fib.
Round 2
Reviewer 3 Report
Comments and Suggestions for Authors
I have no further recommendation.